# Maternal COVID-19 infection and associated factors: A cross-sectional study

**Mwansa Ketty Lubeya**[1,2,3,4]* *, **Jane Chanda Kabwe**[4,5]*, **Moses Mukosha**[2,4,6]‡, **Selia Ng'anjo Phiri**[1,3], **Christabel Chigwe Phiri**[4,7], **Malungo Muyovwe**[7], **Joan T Price**[8,9], **Choolwe Jacobs**[10], **Patrick Kaonga**[10,11]‡

**1** Department of Obstetrics and Gynaecology, School of Medicine, The University of Zambia, Lusaka, Zambia, **2** School of Public Health, Faculty of Health Sciences, University of the Witwatersrand, Johannesburg, South Africa, **3** Women and Newborn Hospital, University Teaching Hospitals, Lusaka, Zambia, **4** Young Emerging Scientists, Lusaka, Zambia, **5** Department of Anaesthesia and Critical Care, National Heart Hospital, Chongwe, Zambia, **6** Department of Pharmacy, School of Health Sciences, The University of Zambia, Lusaka, Zambia, **7** Department of Internal Medicine, Levy Mwanawasa University Teaching Hospital, Lusaka, Zambia, **8** University of North Carolina Global Projects – Zambia, LLC, Lusaka, Zambia, **9** Department of Obstetrics and Gynecology, University of North Carolina School of Medicine, Chapel Hill, North Carolina, United States of America, **10** Department of Epidemiology and Biostatistics, School of Public Health, University of Zambia, Lusaka, Zambia, **11** Department of Bioethics, Bloomberg School of Public Health, Johns Hopkins University, Baltimore, Maryland, United States of America

☯ These authors contributed equally to this work.
‡ MM and PK also contributed equally to this work.
* Ketty.lubeya@unza.zm

**Data Availability Statement:** Data set available on figshare: https://doi.org/10.6084/m9.figshare.21671588.

## Abstract

### Background

Since the declaration of COVID-19 as a global pandemic, several studies have been conducted to examine associated factors. However, few studies have focused on pregnant women infected with COVID-19 in sub-Saharan Africa. Therefore, this study investigated the prevalence and factors associated with COVID-19 infection among pregnant women at the Levy Mwanawasa University Teaching Hospital and Women and Newborn Hospital of the University Teaching Hospitals in Lusaka, Zambia.

### Methods

A cross-sectional study was conducted between March and July 2021. Women were recruited as they presented for antenatal care. Data was collected using a structured questionnaire to capture variables of interest (socio-demographic, clinical and obstetric). COVID-19 diagnosis was made using a nasopharyngeal swab by PCR test. Multivariable logistic regression was used to control for confounding and calculate the odds ratios for each explanatory variable and respective 95% confidence intervals.

### Results

The study enrolled 352 participants with a mean (standard deviation [SD]) age of 30.1 years (5.6). One hundred thirty of 352 (36.9%; 95% CI: 31.9 to 42.2) participants had a confirmed positive SARS-CoV-2 test result. At univariable analysis, factors associated with COVID-19

**Funding:** The authors received no specific funding for this work.

**Competing interests:** The authors have declared that no competing interests exist.

were increased gestational age, education status and maternal HIV serostatus. Women with a secondary level of education were less likely to have COVID-19 infection than those with a primary level of education (AOR = 0.23, 95% CI: 0.09–0.63). On the other hand, a one-week increase in gestational age was associated with higher odds of COVID-19 infection (AOR = 1.03, 95% CI: 1.01–1.06).

## Conclusion

The results showed that the prevalence of COVID-19 infection among pregnant women was 36.9% and was associated with increased gestational age and a lower level of education. To mitigate adverse maternal outcomes, there is a need to screen for COVID-19 strictly and broadly monitor prenatal women presenting for healthcare.

## Introduction

Coronavirus disease 2019 (COVID-19), caused by the Severe Acute Respiratory Syndrome Coronavirus 2 (SARS-CoV-2), was identified in December 2019 and has resulted in many deaths [1]. The illness typically presents with fever, myalgia, shortness of breath and cough [2]. COVID-19 infection began as an epidemic in China and rapidly spread globally, creating stress on public health systems. Therefore, it was declared a pandemic public health emergency by the World Health Organisation (WHO) in March 2020 [3].

In the extant literature, the main factors associated with severe COVID-19 in pregnant women include increasing age, high body mass index, chronic hypertension, pre-eclampsia, and preexisting diabetes [4]. Since the early days of the pandemic, there has been conflicting literature regarding the risk of COVID-19 in pregnant women. While some studies done in China and Senegal do not suggest an increased risk of severe disease among pregnant women compared to the general population [5, 6], another study in France suggested increased respiratory morbidity related to COVID-19 infection [7]. Further, in a multinational cohort study, COVID-19 in pregnancy was associated with increased severe maternal morbidity and mortality compared to pregnant women without COVID-19, which was more pronounced in low-resource settings [8]. Physiological and immunological changes in pregnancy render women susceptible to respiratory infections [5]. It has been shown that during pregnancy, there is a shift from T helper cells 1 to T helper 2 cells, decreased number of circulating natural killer cells, and overexpression of angiotensin-converting enzyme (ACE) receptors [9]. Essentially, these changes occur in the first and third trimesters of pregnancy, making it difficult for pregnant women to fight viral infections [10, 11]. Symptomatic COVID-19 in pregnant women is associated with adverse neonatal outcomes such as increased preterm births, stillbirths and fetal distress compared to non-infected women [12].

There is a lack of data on the prevalence and factors associated with COVID-19 infection among pregnant women, especially in low-resource settings owing to the non-availability of universal screening programs. The few studies done on prevalence in such settings have shown inconsistent results. There seem to be discrepancies with some high-income countries showing lower prevalence compared to low-income countries [12, 13]. Therefore, there is an urgent need to gather more evidence on the prevalence and risk factors of COVID-19 among pregnant women to enhance their care.

Zambia, like many countries, introduced strategies to curb the spread of COVID-19 diseases, such as lockdown, routine hand washing, social distancing and routine screening for any person presenting to a health facility regardless of the presenting symptoms [14]. Therefore, all pregnant women also underwent routine screening for COVID-19 when they presented for antenatal care and delivery at designated health facilities. However, at the time of writing this paper, there was no data on the prevalence and risk factors of COVID-19 among pregnant women in Zambia. Therefore, this study was conducted to determine the prevalence and factors associated with COVID-19 infection in pregnant women who presented to the Levy Mwanawasa University Teaching Hospital (LMUTH) and Women and Newborn Hospital-University Teaching Hospitals (WNH-UTH) in Lusaka, Zambia. These tertiary hospitals served as COVID-19 centres for pregnant and postnatal women in Lusaka province [15]. Data from this study may help public health agencies and clinical care providers identify pregnant women at increased risk infection and to formulate appropriate management strategies.

## Materials and methods

### Study design and setting

This was a cross-sectional study conducted between March and July 2021 among pregnant women attending antenatal care clinics at the LMUTH and WNH-UTH in Lusaka urban Zambia. The WNH-UTH is the largest referral centre in Zambia for obstetrics and gynaecology. On average, 9000 pregnant women are seen annually, and about 4000 births per year are recorded at WNH-UTH. In addition, the hospital receives referrals from over 20 clinics and five general hospitals from surrounding areas of Lusaka and other parts of the country [16]. The LMUTH is a tertiary-level hospital and teaching hospital receiving referrals from facilities located on the Northern and Eastern sides of the district. On average, 8000 women are seen annually and about 6000 deliveries are conducted per year. However, during the study period, there were an average of 1,600 visits by antenatal and postnatal women. New visits were about 100 women a month, while revisits amounted to about 1,100 each month. Postnatal women were about 450 visits per month. Women who had already been enrolled in the study were not eligible for re-enrollment. A detailed description of the study sites has been reported elsewhere [15]. The sites served as COVID-19 centres for both public and private facilities.

### Sample size justification

The sample size for this study was determined using the single population proportion formula considering the assumptions: we assumed the proportion of women seeking ANC with COVID-19 would approximate 50% since there was no previous study in the study setting. We set the level of significance to 5%, 95% confidence intervals, and 5% margin of error, resulting in a sample size of 347 participants. Assuming 10% non-response, the total sample size required for this study was 385.

### Enrolment procedure

All women older than 18 years who presented to the WNH-UTH and LMUTH between March—July 2021 for antenatal care were included subject to consent. During this period, COVID-19 tests were routinely done on all women presenting for antenatal care (ANC) or admission. Women were informed about the rationale of, the mandatory nasopharyngeal swabbing for clients seeking services at the two health facilities. Therefore, women who did not undergo COVID-19 testing or tested more than ten days before presentation were excluded from the study.

The nasopharyngeal swabs, and laboratory testing was free in these two facilities. Total enumeration was done for women who presented in the study period, subject to eligibility criteria being met. Women who had already been enrolled in the study and came for revisits were not eligible for re-enrollment.

Potential clients were approached by trained research assistants and informed about the study. If they were willing to participate, information sheets were shared, and subsequently signed informed consent. This study was conducted while adhering to social distancing recommendations and applicable WNH-UTH policies regarding COVID-19 prevention. Full personal protective equipment was available and was used appropriately by the research team. In addition, the research team provided masks to potential clients who did not have them at the time of recruitment.

## Data collection tool

A structured data collecting tool developed based on similar studies was piloted with 10% of the calculated sample size at a different hospital and the data is not included in the analysis. Face validity of the tool was deduced using cognitive interviews with a subsample of 10 pregnant women to assess whether the meaning of the questions was clear. For the reliability test, items to be returned in the tool assessment was conducted based on item-to-total and inter-item correlations. The absolute reliability was assessed using Cronbach's alpha, and a scale of 0.7 coefficient was considered acceptable [17]. The outcome variable COVID-19 infection was measured on a binary scale (no = 0, yes = 1). The test for SARS-CoV-2 was performed as part of the routine standard of care by trained laboratory scientist using Polymerase Chain Reaction (Aptima. SARA-CoV-2 Assay or applied biosystems by Thermo Fisher Scientific-USA/ The Netherlands). The independent variables included socio-demographic, clinical and obstetric characteristics. These included maternal age (age at last birthday), parity (number of times a woman has given birth to a foetus with gestational age at or above 28 weeks); gravidity (i.e., the number of previous pregnancies, including the present one and miscarriages and ectopic pregnancies), gestational age (as number of completed weeks of gestation based on last menstrual period), ANC visits (once, twice, three times, more than three times), BMI was calculated from the current visit's weight and height. Based on WHO criteria, BMI was divided into three groups, thin or healthy (BMI < 25 kg/m$^2$), overweight (25 < BMI < 29 kg/m$^2$) and obese (BMI $\geq$ 30 kg/m$^2$), taking into account the challenges of measuring BMI in pregnancy, marital status, education level (none, primary, secondary, tertiary), Employed (yes, no); Pre-eclampsia (yes, no), chronic hypertension (yes, no), HIV status (positive, negative).

## Statistical analysis

Data was processed and cleaned to minimize entry errors and identify outliers and missing values. We conducted a descriptive analysis using frequencies and percentages for categorical variables. After testing for assumptions of normal distribution (using Shapiro-Wilk test and graphically Q-Q plots), we reported continuous variables as mean (SD) or median (IQR) as appropriate. Fishers exact or Pearson Chi-square tests were done to identify statistically significant associations between COVID-19 infection and categorical explanatory variables. On the other hand, Student T-test or Wilcoxon Rank sum test was used to assess mean or median differences for continuous variables. Finally, multivariable logistic regression was used to control for confounding and obtain the odds ratios for each explanatory variable and respective 95% confidence intervals and p-values. All univariable logistic regression model variables with a p-value <0.2 were added to the multivariable logistic regression model. An investigator led stepwise regression techniques with liberal p-value (p<0.15) for exclusion was used for the

multivariable model. In the final model interactions between significant variables were investigated and none reached statistical significance. Only variables with a p-value less than 0.05 were considered statistically significant at 95% confidence intervals in the multivariable logistic regression model. We used Stata/IC version 16.1 (Stata Corporation, Texas, TX, USA) for statistical analysis.

### Ethical considerations

The ethical approval was obtained from the University of Zambia Biomedical Ethics Research Committee (UNZABREC), reference number: UNZA-1187/2020. Additional permission was granted from the National Health Research Authority (NHRA) and the senior medical superintendents at the WNH-UTH & LMUTH. Participant's personal information was de-identified, and stored in protected files and locked cabinets. Participants were recruited once their COVID-19 result status had been given to them by their primary healthcare teams. Consenting participants were told about withdrawing from the study without impacting their care if they became overwhelmed with emotions due to the diagnosis of COVID-19. Participants who needed psychological care were referred to the psychiatry clinic domiciled at the University Teaching Hospitals in Lusaka, Zambia.

## Results

### Socio-demographics, obstetric and clinical characteristics

We screened 385 participants, with 33 not meeting the eligibility criteria. The final analysis included 352 participants, of whom 130 (36.9%, 95% CI: 31.9% to 42.2%) had a confirmed positive SARS-CoV-2 test result. The overall mean (standard deviation [SD]) age of study participants was 30.1 years (5.6). In addition, the median (interquartile range [IQR]) were parity 2 (1–3), gravidity 3 (2–4) and gestational age in weeks 30 (20–36). Slightly over half, 181 (51.4%) attained a tertiary level of education. About 13(3.7%) had chronic hypertension, 32 (9.1%) were HIV seropositive, 315(89.5%) were married, and 14(4.0%) had a diagnosis of pre-eclampsia. Additionally, about two-thirds of the participants had visited antenatal care more than three times. Furthermore, about half, 62(17.6%), of the participants had a BMI of 25–30, and 174 (49.4%) were employed (Table 1).

### Factors associated with COVID-19 among pregnant women

Table 2 shows results from a univariable and multivariable regression analysis of factors associated with COVID-19. At univariable analysis, factors associated with COVID-19 were increased gestational age, education status and maternal HIV status. Only the education status remained significantly associated with COVID-19 infections in the multivariable analysis and was more pronounced than in the univariable analysis. Education status (i.e., secondary compared to primary AOR = 0.23, 95% CI: 0.09–0.63) was associated with 77% lower odds of having COVID-19 infection. Conversely, the effect of a one week increase in gestational age on COVID-19 infections remained constant at 3%.

## Discussion

We sought to understand prevalence and characteristics of pregnant women diagnosed with COVID-19 in Zambia. We found a prevalence of COVID-19 of 36.9% among pregnant women presenting to the two health facilities. Women with a secondary level of education were less likely to have COVID-19 infection than those with primary level of education, and a one week increase in gestational age was associated with higher odds of COVID-19.

**Table 1. Demographic, obstetric and clinical characteristics of study participants by COVID-19 status, N = 352.**

| Variable | Level | Total population n(%)/Mean(SD)/Median (IQR) | COVID-19[f] | | P-value |
|---|---|---|---|---|---|
| | | | No, n (%)/mean(SD)/Median (IQR) | Yes, n (%)/Mean(SD)/Median (IQR) | |
| Maternal age | Years | 30.1(5.6) | 30.2(5.8) | 29.9(5.4) | 0.698[a] |
| Parity | Count | 2(1–3) | 1(1–2) | 2(1–3) | 0.121[b] |
| Gravidity | Count | 3(2–4) | 2.5(2–3) | 3(1–4) | 0.577[b] |
| Gestational age | Week | 30(20–36) | 28(18–36) | 31.5(24–35) | 0.097[b] |
| ANC visits, n = 328 | <3 | 105(32.0) | 76(33.2) | 29(27.4) | 0.376[c] |
| | ≥3 | 223(68.0) | 146(65.8) | 77(72.6) | |
| BMI (kg/m2)[e], n = 132 | <25 | 58(43.9) | 46(42.6) | 12(50.0) | 0.498[d] |
| | 25–29 | 62(47.0) | 53(49.1) | 9(37.5) | |
| | ≥30 | 12(9.1) | 9(8.3) | 3(12.5) | |
| Marital status, n = 350 | Single | 35(10.0) | 16(7.2) | 19(14.8) | 0.022[c] |
| | Married | 315(89.0) | 206(92.8) | 109(85.2) | |
| Education, n = 347 | Primary | 37(10.7) | 17(7.7) | 20(16.0) | 0.001[d] |
| | Secondary | 129(37.2) | 85(38.3) | 44(35.2) | |
| | Tertiary | 181(52.2) | 120(54.5) | 61(48.8) | |
| Employed, n-348 | No | 173(49.9) | 103(46.4) | 70(56.0) | 0.086[c] |
| | yes | 174(49.4) | 119(53.6) | 55(44.0) | |
| Pre-eclampsia | No | 338(96.0) | 208(93.7) | 130(100) | 0.003[d] |
| | Yes | 14(4.0) | 14(6.3) | - | |
| Chronic hypertension | No | 339(96.3) | 211(95.1) | 128(98.5) | 0.144[d] |
| | Yes | 13(3.7) | 11(4.9) | 2(1.5) | |
| Maternal HIV serostatus | Negative | 320(90.9) | 209(94.1) | 111(85.4) | 0.006[c] |
| | Positive | 32(9.1) | 13(5.9) | 19(14.6) | |

[a]Student T-test,

[b]Wilcoxon ranksum test,

[c]Chi-square test,

[d]Fishers exact test,

[e]BMI categories were done based on the WHO criteria,

[f]COVID-19 positive test result was confirmed by polymerase chain reaction,

ANC- Antenatal care, BMI-body mass index, HIV-human immunodeficiency virus, all variables with missing values have been indicated with respective "n" used for complete case analysis.

The prevalence of COVID-19 infection among pregnant women is variable in the extant literature with most studies having been conducted in high-income countries. A hospital in Houston, USA found a prevalence of 8.0% [18], another hospital in Connecticut, USA reported prevalence of 3.9% [19], and 3.4% in Jammu and Kashmir, India [20]. These findings are all much lower than what we found, plausible explanation is that we conducted our study during the peak of wave three of the COVID-19 pandemic which was the worst experienced by Zambia [14]. Further, the two sites served as COVID-19 centres and catered for women from all levels of care from both government and private facilities, and additionally referrals from other parts of the country. Women with a positive COVID-19 test from any of the health centres were referred to WNH-UTH or LMUTH for further assessment. Further, studies have shown that people of colour are at an increased risk for COVID-19 infection [21–23], which could partly explain the high prevalence in this setting.

**Table 2. Multivariable logistic regression of factors associated with COVID-19 infection among pregnant women.**

| Variable | Level | Crude OR (95% CI) | Adjusted OR (95% CI) |
|---|---|:---:|:---:|
| Maternal age (years) | Unit increase | 1.01(0.95–1.03) | 1.01(0.94–1.07) |
| Parity | Unit increase | 1.12(0.96–1.31) | 0.97(0.74–1.27) |
| Gestational age (weeks) | Unit increase | **1.03 (1.01–1.06)** | **1.03(1.01–1.06)** |
| Marital status | Unmarried | Ref | Ref |
|  | Married | **0.45(0.22–0.91)** | 0.45(0.15–1.38) |
| Education level | Primary | Ref | Ref |
|  | Secondary | **0.44(0.21–0.92)** | **0.23(0.09–0.63)** |
|  | Tertiary | **0.43(0.21–0.88)** | 0.37(0.13–1.07) |
| Employed | No | Ref | Ref |
|  | yes | 0.68(0.44–1.06) | 0.80(0.40–1.61) |
| Chronic hypertension | No | Ref | |
|  | Yes | 0.30(0.07–1.37) | |
| Maternal HIV serostatus | Negative | Ref | Ref |
|  | Positive | **2.75(1.31–5.78)** | 1.48(0.54–4.04) |

Key: boldface indicates statistical significance (p<0.05), the multivariable model was fitted using COVID-19 as a binary variable (yes = 1, no = 0) and age set as a priori, adjustment variables (maternal age, parity, gestational age, marital status, education level, employment status and maternal HIV status), Chronic hypertension was dropped from final model due to collinearity.

We found that the level of education was significantly associated with COVID-19 infection. is. Pregnant women with secondary education were less likely to present with COVID-19 than women with primary school education. This is similar to a study done in Europe which suggested that people of lower education are economically disadvantaged in terms of housing and live in overcrowded spaces, making it difficult to social distance thus increasing their risk of COVID-19 [24]. Furthermore, pregnant women with low education may have access to knowledge on infection prevention but have poor understanding and attitudes towards practicing preventive measures [25].

In our study, gestational age was significantly associated with COVID-19 infection. The finding is similar to other literature which showed that most women presented with the disease in the third trimester of pregnancy [11]. This could be probably due to the immunological changes that take place as a woman approaches term to a pro inflammatory state [26].

Our study found that being married was a predisposing factor in the univariable analysis. However, it was no longer significant after adjusting for confounders. A mixed-methods study done in Lima found that pregnant women who were cohabiting had lower chances of having COVID-19 when compared with other marital status [27]. Most women included in this study were married, so we could not detect the true effect. Furthermore, married women may be at risk of infection from their partners, effectively increasing contact with others. This finding provides preliminary evidence in our setting, but more studies on this association are needed.

In this cross-sectional analysis, we found that pregnant women with HIV had higher odds of COVID-19 in the univariate analysis but it was no longer significant in the adjusted model. Similarly, pregnant women living with HIV were not at higher risk of COVID-19 compared to women without HIV in a study by De Waard et al. in South Africa [28]. However, some studies argue that having HIV and being on a Tenofovir-based treatment regimen could protect against COVID-19, although more research is recommended in high burden geographical regions.

Our results showed a trend toward women 30 years or younger (53.4%) having a higher prevalence of COVID-19 infection [29], although not statistically significant. However, other similar studies have reported a significant association. For example, a study done in Italy reported that women less than 35 years old had a higher risk of infection than older women [30]. One plausible reason could be due to more regularly mild upper respiratory infections experienced by younger individuals leading to immune cross-protection during contact with an individual with COVID-19 infection [31].

This study has some limitations. The study was conducted at two tertiary hospitals in the capital city of Zambia and may not be generalisable to the general population. However, these two centres were provincial COVID-19 centres admitting all pregnant women with COVID-19 in the province, which harboured the epicentre for COVID-19 in the country. Further, due to the cross-sectional nature, there was no follow-up after the once-off interaction with the participants making it difficult to understand other factors that might have been missed at time of the interview, including pregnancy outcomes. The strengths of the study are that it is the first of its kind conducted in this setting, focusing on a special population, hence is an excellent step in understanding the factors surrounding the disease and a possible guide to mitigation steps. Further, the diagnosis of COVID-19 was laboratory based, removing inherent errors of a clinical diagnosis. Finally, the study was done in a set-up with universal screening for COVID-19.

## Conclusion

The prevalence of COVID-19 among pregnant women was 36.9% and significantly associated with later gestational age and low education level. As such, clinicians and public health personnel should pay special attention to this group to reduce the morbidity and mortality associated COVID-19.

## Acknowledgments

We would like to acknowledge the following research assistants: Ntungo Siulapwa, Kasukula Nathaniel Kaunda, Theresa Shema Nzayinsenga, Anthony Limbumbu, James Nyirenda and Mable Ndambo. Further, we would to thank Everlyn Chirwa and Faydes Malupande Banda, the Nurses in charge of the maternal COVID-19 clinics/wards at the time of data collection. We would also like to thank Caren Chizuni, Safe Motherhood Officer at Zambia's Ministry of Health Headquarters for the technical support during the study period.

## Author Contributions

**Conceptualization:** Mwansa Ketty Lubeya, Jane Chanda Kabwe, Patrick Kaonga.

**Data curation:** Moses Mukosha, Joan T Price.

**Formal analysis:** Moses Mukosha, Patrick Kaonga.

**Investigation:** Mwansa Ketty Lubeya, Jane Chanda Kabwe, Christabel Chigwe Phiri, Malungo Muyovwe, Choolwe Jacobs.

**Methodology:** Mwansa Ketty Lubeya, Jane Chanda Kabwe, Moses Mukosha, Selia Ng'anjo Phiri, Christabel Chigwe Phiri, Malungo Muyovwe, Joan T Price, Choolwe Jacobs, Patrick Kaonga.

**Supervision:** Mwansa Ketty Lubeya, Jane Chanda Kabwe.

**Visualization:** Joan T Price.

**Writing – original draft:** Mwansa Ketty Lubeya, Jane Chanda Kabwe, Moses Mukosha, Selia Ng'anjo Phiri, Christabel Chigwe Phiri, Patrick Kaonga.

**Writing – review & editing:** Mwansa Ketty Lubeya, Jane Chanda Kabwe, Moses Mukosha, Selia Ng'anjo Phiri, Christabel Chigwe Phiri, Malungo Muyovwe, Joan T Price, Choolwe Jacobs.

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
