## [Decision Letter · Decision Letter 0]

31 Oct 2022

PONE-D-22-21448Maternal COVID-19 Infection and Associated Factors: A Cross-Sectional StudyPLOS ONE

Dear Dr. Lubeya,

Thank you for submitting your manuscript to PLOS ONE. After careful consideration, we feel that it has merit but does not fully meet PLOS ONE’s publication criteria as it currently stands. Therefore, we invite you to submit a revised version of the manuscript that addresses the points raised during the review process.

ACADEMIC EDITOR: Dear Authors,In addition to the reviewers' comments, may you ensure that you utilize the appropriate sign symbol of PLOS ONE in your Tables. Please see the authors' guide. You may report the univariable regression results alongside the multivariable if it is relevant to your presentation ============================== Please submit your revised manuscript by Dec 15 2022 11:59PM. If you will need more time than this to complete your revisions, please reply to this message or contact the journal office at plosone@plos.org. Please include the following items when submitting your revised manuscript:A rebuttal letter that responds to each point raised by the academic editor and reviewer(s). You should upload this letter as a separate file labeled 'Response to Reviewers'.A marked-up copy of your manuscript that highlights changes made to the original version. You should upload this as a separate file labeled 'Revised Manuscript with Track Changes'.An unmarked version of your revised paper without tracked changes. You should upload this as a separate file labeled 'Manuscript'.

We look forward to receiving your revised manuscript.

Kind regards,

Gbenga Olorunfemi, MBBS,MSC,FMCOG,FWASC

Academic Editor

PLOS ONE

Journal Requirements:

2.  You indicated that you had ethical approval for your study. Please clarify whether minors (participants under the age of 18 years) were included in this study. If yes, in your Methods section, please ensure you have also stated whether you obtained consent from parents or guardians of the minors included in the study or whether the research ethics committee or IRB specifically waived the need for their consent.

“MKL& MM would like to acknowledge that some of their time is supported by the UNC-UNZA-Wits Partnership for HIV and Women’s Reproductive Health (Grant number: D43 TW010558). MKL & CCP would like to acknowledge that some of their time is supported by the TESA III Project funded by EDCTP (Grant number: CSA2020NoE-3104). JTP is supported by a career development grant from the US National Institutes of Health (K01 TW010857). We would further like to acknowledge the following research assistants: Ntungo Siulapwa, Kasukula Nathaniel Kaunda, Theresa Shema Nzayinsenga, Anthony Limbumbu, James Nyirenda and Mable Ndambo. Further, we would to thank Everlyn Chirwa and FaydesMalupande Banda, the Nurses in charge of the maternal COVID-19 clinics/wardS at the time of data collection We would also like to thank Caren Chizuni, Safe Motherhood Officer at Zambia’s Ministry of Health Headquarters for the technical support during the study period.”

“This study did not receive any funding”

“This study did not receive any funding”

“No authors have a competing interest”

6. We note that you have indicated that data from this study are available upon request. PLOS only allows data to be available upon request if there are legal or ethical restrictions on sharing data publicly. For more information on unacceptable data access restrictions, please see http://journals.plos.org/plosone/s/data-availability#loc-unacceptable-data-access-restrictions.

Reviewers' comments:

Reviewer's Responses to Questions

**Comments to the Author**

1. Is the manuscript technically sound, and do the data support the conclusions?

Reviewer #1: Yes

Reviewer #2: Partly

2. Has the statistical analysis been performed appropriately and rigorously? 

Reviewer #1: Yes

Reviewer #2: No

3. Have the authors made all data underlying the findings in their manuscript fully available?

Reviewer #1: No

Reviewer #2: No

4. Is the manuscript presented in an intelligible fashion and written in standard English?

Reviewer #1: No

Reviewer #2: Yes

5. Review Comments to the Author

Reviewer #1: This is an interesting study this study aimed at investigating the prevalence

and factors associated with COVID-19 infection among pregnant women at the Levy Mwanawasa University Teaching Hospital and Women and Newborn Hospital of the University Teaching Hospitals in Lusaka, Zambia.

There are a number of issues:

ABSTRACT AND METHODOLOGY

The authors stated: COVID-19 diagnosis was made using a nasopharyngeal swab by PCR test.

The authors should clarify how the test was done either routinely or voluntarily? What is the testing protocols in the study hospitals? How long does the test take to come out? Who paid for the COVID-19 test?

How was the sampling technique achieved in the study?

RESULTS

The authors should begin by stating how many participants were assessed for eligibility and how many were recruited, excluded and reason for exclusion.

Reviewer #2: Reviewer comments

General comment

The tittle is interesting. I appreciate your view. As we know COVID -19 is a global pandemic and pregnant women have more risk than other population, Also no clinical trial was done for pregnant mothers, so knowing risk factors for COVID-19 is important for designing and implementing effective prevention strategies.

Specific comments

I try to address my comments part by part

Introduction

1. In the introduction section paragraph 2. You said “Since the early days of the pandemic, there has been conflicting literature regarding the risk of COVID-19 in pregnant women [5-7]”. What does it mean? I think all your cited article showed that pregnant women are more venerable for COVID-19 than the general population. What was the controversy idea?

Sample size calculation

1. Your sample is not correct. Based on your assumption your sample size 384 (before adding non response rate). After considering 10% of non-response rate the final sample size should be 422. Please check it.

Enrolment procedure

1. As you said in WNH-UTH and LMUTH on average, 9000 and 8000 pregnant women seen annually, also your study period was March and July 2021 (almost 5 months). Accordingly, approximately around 7000 pregnant women visited this two hospitals with in your study period, On the other hand your sample size is 385. How can you draw your sample? Please say something about your sampling technique (how the actual sample was drawn?

2. How do you manage if the women visited the hospitals for more than once during your study period?

Data collection tool

1. What is the difference between pilot study and pretest? Which one is included in the analysis?

Results

1. Some results not give a total of 352 study participant. For example ANC visit, BMI, marital status, education, employed.

2. Delete univariable analysis OR result from the text, it is enough to be presented in the table.

3. What is the final result (multivariable logistic regression) of chronic hypertension?

Discussion

Further, studies have shown that people of colour are at an increased risk for COVID-19 infection which could explain the high prevalence we found at this setting. If there is any evidence cite?

6. PLOS authors have the option to publish the peer review history of their article (what does this mean?). If published, this will include your full peer review and any attached files.

Reviewer #1: **Yes: **George Eleje

Reviewer #2: No

---

## [Author Response · Author response to Decision Letter 0]

13 Dec 2022

6th December, 2022.

The Academic Editor,

PLOS ONE Journal,

Dear Dr Gbenga Olorunfemi,

RE: Manuscript ID: PONE-D-22-21448__ Title: Maternal COVID-19 Infection and Associated Factors: A Cross-Sectional Study

We are sincerely greatful that you invited us to submit a revised version of the manuscript that addresses the points raised during the review process. We took all the points and comments raised with the seriousness they deserve as we believe this would help in improving the manuscript achieve the publication criteria. The response is in the table is the point-by point rebuttal under attached documents. 

Thank you once again for taking your time to review our manuscript.

---

## [Decision Letter · Decision Letter 1]

24 Jan 2023

Maternal COVID-19 Infection and Associated Factors: A Cross-Sectional Study

PONE-D-22-21448R1

Dear Dr. Lubeya,

We’re pleased to inform you that your manuscript has been judged scientifically suitable for publication and will be formally accepted for publication once it meets all outstanding technical requirements.

Kind regards,

Gbenga Olorunfemi, MBBS,MSC,FMCOG,FWASC

Academic Editor

PLOS ONE

Additional Editor Comments (optional):

Reviewers' comments:

Reviewer's Responses to Questions

**Comments to the Author**

1. If the authors have adequately addressed your comments raised in a previous round of review and you feel that this manuscript is now acceptable for publication, you may indicate that here to bypass the “Comments to the Author” section, enter your conflict of interest statement in the “Confidential to Editor” section, and submit your "Accept" recommendation.

Reviewer #1: All comments have been addressed

2. Is the manuscript technically sound, and do the data support the conclusions?

Reviewer #1: Partly

3. Has the statistical analysis been performed appropriately and rigorously? 

Reviewer #1: Yes

4. Have the authors made all data underlying the findings in their manuscript fully available?

Reviewer #1: Yes

5. Is the manuscript presented in an intelligible fashion and written in standard English?

Reviewer #1: Yes

6. Review Comments to the Author

Reviewer #1: The authors have addressed the comments raised in the earlier review. The manuscript now has better information.

7. PLOS authors have the option to publish the peer review history of their article (what does this mean?). If published, this will include your full peer review and any attached files.

Reviewer #1: **Yes: **George Eleje

---

## [Editor Report · Acceptance letter]

30 Jan 2023

PONE-D-22-21448R1 

Maternal COVID-19 Infection and Associated Factors: A Cross-Sectional Study 

Dear Dr. Lubeya:

I'm pleased to inform you that your manuscript has been deemed suitable for publication in PLOS ONE. Congratulations! Your manuscript is now with our production department. 

Kind regards, 

on behalf of

Dr. Gbenga Olorunfemi 

Academic Editor

PLOS ONE